# Clinical Characteristics of *Actinomyces viscosus* Bacteremia

**DOI:** 10.3390/medicina57101064

**Published:** 2021-10-05

**Authors:** Yi-Chun Hsiao, Yi-Hsuan Lee, Chun-Mei Ho, Chien-Hao Tseng, Jui-Hsing Wang

**Affiliations:** 1Department of Internal Medicine, Division of Infectious Diseases, Taichung Veterans General Hospital, Taichung 40705, Taiwan; hetairoiyig@gmail.com (Y.-C.H.); wenily@vghtc.gov.tw (C.-M.H.); tedi3tedi3@vghtc.gov.tw (C.-H.T.); 2Department of Post-Baccalaureate Veterinary Medicine, Asia University, Taichung 41354, Taiwan; joyce0936@gmail.com; 3Department of Internal Medicine, Division of Infectious Disease, Taichung Tzu Chi Hospital, Buddhist Tzu Chi Medical Foundation, Taichung 40705, Taiwan; 4Department of Internal Medicine, School of Medicine, Buddhist Tzu Chi Medical Foundation Taichung Tzu Chi Hospital, Taichung 427213, Taiwan

**Keywords:** *Actinomyces viscosus*, *Actinomyces*, actinomycosis, cellulitis, bacteremia, gingivitis, sequence analysis

## Abstract

*Background and Objectives*: *Actinomyces* species are part of the normal flora of humans and rarely cause disease. It is an uncommon cause of disease in humans. The clinical features of actinomycosis have been described, and various anatomical sites (such as face, bones and joints, respiratory tract, genitourinary tract, digestive tract, central nervous system, skin, and soft tissue structures) can be affected. It is not easy to identify actinomycosis because it sometimes mimics cancer due to under-recognition. As new diagnostic methods have been applied, *Actinomyces* can now more easily be identified at the species level. Recent studies have also highlighted differences among *Actinomyces* species. We report a case of *Actinomyces viscosus* bacteremia with cutaneous actinomycosis. *Materials and Methods*: A 66 years old male developed fever for a day with progressive right lower-leg erythematous swelling. Blood culture isolates yielded *Actinomyces* species, which was identified as *Actinomyces viscosus* by sequencing of the 16S rRNA gene. In addition, we searched for the term *Actinomyces* or actinomycosis cross-referenced with bacteremia or “blood culture” or “blood stream” from January 2010 to July 2020. The infectious diseases caused by species of *A. viscosus* from January 1977 to July 2020 were also reviewed. *Results*: The patient recovered well after intravenous ampicillin treatment. Poor oral hygiene was confirmed by dental examination. There were no disease relapses during the following period. Most cases of actinomycosis can be treated with penicillin. However, clinical alertness, risk factor evaluation, and identification of *Actinomyces* species can prevent inappropriate antibiotic or intervention. We also compiled a total of 18 cases of *Actinomyces* bacteremia after conducting an online database search. *Conclusions*: In summary, we describe a case of fever and progressive cellulitis. *Actinomyces* species was isolated from blood culture, which was further identified as *Actinomyces viscosus* by 16S rRNA sequencing. The cellulitis improved after pathogen-directed antibiotics. Evaluation of risk factors in patients with *Actinomyces* bacteremia and further identification of the *Actinomyces* species are recommended for successful treatment.

## 1. Introduction

The first case of actinomycosis was described in cattle by a pathologist in 1877 [1]. Then, shortly afterward in 1890, *Actinomyces israelii* was discovered in humans [2]. These bacteria are Gram-positive, filamentous, and rod-shaped. They are mostly facultative anaerobic organisms, which normally colonize the oral cavity, gastrointestinal tract, colon, and vagina [3,4]. However, only a handful of species are known to be associated with human infections such as *A. israelii*, *A. meyeri*, *A. neuii*, and *A. turicensis* [3,4]. *Odontomyces viscosus* was first found in the periodontal plaque of hamsters in 1958. The species was classified and named “*Actinomyces viscosus*” [5,6]. The two closely related species *A. viscosus* and *A. naeslundii* showed high phenotypic and serological relatedness and could be isolated from dental plaque and mucosa samples [7]. *A. naeslundii* Serotype II was renamed “*Actinomyces oris*” in 2009 [8], and human strains which have been assigned to *A. viscosus* are likely members of *A. oris* [9]. Despite being a commensal bacterium in the majority of human adults with teeth, *A. viscosus* has rarely been reported to cause disease [10].

*Actinomyces* was redefined to include catalase-positive organisms in 1969 [11]. Unlike other species which are catalase-negative and indole-positive, these colonies are catalase-positive and indole-negative. *Actinomyces* species can be cultured on blood agar (BA) with selective nutritional requirements [12]. In the 1980s, a new standard of 16S rRNA gene sequencing for bacterial identification was introduced and also used to establish the genotypic taxonomy [13,14,15]. In order to identify bacteria at the genus and species level, we used 16S rRNA gene sequencing technology to quickly perform reliable identification of bacteria in this case.

Evidence of actinomycosis infection is based on accurate identification of *Actinomyces* species. Therefore, accuracy is a key factor in preventing unnecessary invasive intervention and facilitates the selection of proper treatment [16,17]. The most common pathogenic actinomycosis species include *A. viscosus* and *A. meyeri* [2,3,4,12]. According to some reviews, *A. israelii* and *A. meyeri* have been identified as frequently encountered specimens from periappendiceal abscesses and abdominal actinomycosis [12,18].

Cutaneous actinomycosis is uncommon in clinical practice [19] and is usually a secondary infectious process with an underlying focus in deeper tissues [20], or it may appear as a result of hematogenous spread from an actinomycotic lesion elsewhere in the body [4]. In cutaneous actinomycosis, the commonly found causative organisms were *A. meyeri* and *A. viscosus* according to previous reports [4,12]. Oral hygiene is a recognized risk factor for the development of cutaneous actinomycosis [21]. According to a literature review of original clinical studies on *Actinomyces*, this species can be a source of invasive disease when superadded by periodontal disease and poor oral hygiene, leading to the development of infections. Although oral cervicofacial actinomycosis is the common form, *Actinomyc**es* can also result in infection of the thoracic, abdominopelvic, cutaneous, musculoskeletal system, pericardium, and central nervous system, as well as in disseminated disease [3].

In the present review, we report a case with *A. viscosus* bacteremia and cellulitis in a patient with poor oral hygiene, and we gathered this information to provide a comprehensive and microbiologically consistent overview of the *A. viscosus* bacteremia and cutaneous actinomycosis in human infections.

## 2. Materials and Methods

### 2.1. Case Presentation

A 66 years old male with hypertension and bipolar disease under treatment with lithium, fludiazepam, and seroquel came to our emergency department due to fever up to 38.9 °C (102 °F) without chills for a day. The patient complained of an erythematous painful swelling over the right lower leg, which developed gradually during the previous week. The patient also mentioned nausea with vomiting for 2 days, which did not seem to be related to meals. Otherwise, no other symptoms were mentioned. There was no traumatic history, no bug bite, and no exposure to livestock. On physical examination, blood pressure was 131/66 mmHg, heart rate was 91/min, respiratory rate was 18/min, and body temperature was 38.9 ℃. Erythematous change measuring around 10 × 20 cm over the anterior tibial region with tenderness was found, but there were no excoriated skin lesions in this region. Laboratory tests showed notable leukocytosis with neutrophil predominance. C-reactive protein level was 17.9 mg/dL, procalcitonin level was 14.5 ng/mL, serum creatinine level was 1.55 mg/dL, and ClCr level was 51 mL/min. Two sets of blood cultures were obtained and yielded *Actinomyces* sp. Further sensitivity tests showed susceptibility to clindamycin, penicillin, ampicillin–sulbactam, cefoxitin, cefmetazole, and carbapenem, but resistance to metronidazole. Clindamycin 600 mg Q8H was administered for *Actinomyces* bacteremia and cellulitis. The erythematous swelling of the right lower leg improved gradually, and antibiotic treatment was shifted to sole intravenous ampicillin 2 g every 6 h on admission day 10 due to clinical improvement. There was no bacterial growth in the following blood culture.

Due to poor oral hygiene, a dentist was consulted for evaluation. Full-mouth gingivitis with plaque deposition and easy bleeding on probing was found during the dental examination. The patient then completed 7 days of intravenous ampicillin and was discharged with oral ampicillin 500 mg every 6 h. Further 16s rDNA sequencing for identification of pathogens revealed *Actinomyces viscosus*. The primers used for amplification of the 16S ribosomal DNA gene were 27F/1525R, 8F2/806R, and fD1modF/16S1RR-B. The amplification products obtained by PCR were sequenced, and the sequences obtained (791 bp) were compared to known 16S ribosomal DNA sequences in the GenBank database of the National Center for Biotechnology Information using the BLASTN algorithm (http://www.ncbi.nlm.nih.gov/blast accessed on 7 September 2021). The closest match was obtained with *Actinomyces viscosus* (GenBank accession number NR_113030; maximal score 848, E value 0.0, and maximal identity 84% (600/711)). There was no evidence of any diseases during follow-up.

### 2.2. Literature Review

A review of the English-language literature on *Actinomyces* bacteremia was conducted. Key search terms were *Actinomyces* OR actinomycosis cross-referenced with bacteremia OR “blood culture” OR “blood stream” in Pub Med/NCBI and other similar databases from January 2010 to July 2020. The infectious diseases caused by species of *A. viscosus* from January 1977 to July 2020 were also reviewed. All of the relevant information obtained from the literature review is presented in tables.

## 3. Results

We compiled a total of 18 cases of *Actinomyces* bacteremia from the online database search. The clinical and microbiological characteristics of these cases are summarized in Table 1.

The average age of the patients was 52.5 years, ranging widely from 23 weeks gestational age to 90 years old. Eight patients (44%) were male. Seven patients (39%) had an underlying systemic condition, which resulted in a relatively immunocompromised status (one alcoholic liver cirrhosis, four diabetes mellitus, one preterm labor, and one monoclonal gammopathy). Seven patients (39%) had previous exposure to invasive procedures or implantations (two IUD, one TVOR, one colonoscopy, one surgery, one endoscope, and one central-line placement). Eleven (61%) patients had fever. Fifteen (83%) patients had a primary site of infection; five (33% = 5/15) of these were gynecology–genital organ infection (including testicular abscess), followed by three (20%) urinary tract infections, two (13%) pulmonary infections, two (13%) soft-tissue infections, and one case each of (7%) meningitis, abdominal infection, and endocarditis.

Most of the treatments included penicillin-based antibiotic administration. Alternative antibiotics included third-generation cephalosporin (ceftriaxone for example) or clindamycin. Broad-spectrum antibiotics, such as carbapenems, were used in severe or comorbid patients. Three patients (17%) required further surgical intervention (endocarditis, necrotizing soft-tissue infection, and pelvic inflammatory disease with abscess). Most of the patients had favorable outcomes, and only a few patients with multiple comorbidities died or suffered from morbidity.

We also reviewed previous infectious disease with the species *Actinomyces viscosus* as a causative organism from January 1977 to July 2020, using the search term “*Actinomyces viscosus*” OR “*A. viscosus*”, as shown in Table 1B. Fifteen studies were found with a total of 19 relevant human cases. The clinical and microbiological characteristics of these cases are summarized in Table 2. The 19 cases comprised 10 men and nine women. The medium age of the patients was 35 years, ranging in age from gestational age of 23 weeks to 81 years old. Six patients (32%) had a relatively immunocompromised status (multiple myeloma, alcoholism, pancreatic cancer, acute lymphoblastic leukemia under chemotherapy, and psoriatic arthritis under methotrexate). Eleven patients (57%) suffered from fever. As to the acquired specimens, in four patients (21%), *Actinomyces viscosus* was found from blood culture. Seven patients (37%) needed biopsy or tissue culture for *Actinomyces viscosus* isolation. Other specimens included pus, drainage or discharge from submandibular, neck, chest wall, and breast abscess, subdural empyema, pleural fluid, percutaneous transtracheal aspiration, and vitreous washings. Most of the specimens were diagnosed by biochemical analysis, and only two specimens were analyzed using PCR (in the year 2005) or 16s sequencing (in the year 2018). Only one patient suffered from subdural empyema, resulting in death.

## 4. Discussion

*Actinomyces* species are opportunistic pathogens and capable of causing disease, which often invade the body following the disruption of mucosal barriers [4,8].

There are nine Actinomyces species that are commonly found in the oral cavity, namely, A. israelii, A. viscosus, A. odontolyticus, A. naeslundii, A. georgiae, A. gerensceriae, A. meyeri, A. radicidentis, and A. graevenitzii [52,53]. A. israelii is known as the most common bacterium associated with classical actinomycosis, which has been linked with dental abscesses and oral infections [12]. In this study, we reviewed cases of Actinomyces bacteremia in databases and found that 15 of 18 patients had at least one risk factor.

According to the analysis, seven patients were relatively immunocompromised (one alcoholic liver cirrhosis, four diabetes mellitus, one preterm labor, and one monoclonal gammopathy). Seven cases had a procedure-related risk factor (two IUD, one TVOR, one colonoscopy, and one surgery), and four cases had evidence of poor oral hygiene. The diagnostic method and basic characteristics of *Actinomyces* bacteremia cases reported from 2010 January to 2020 July are summarized in Table 1 and Table 2.

A phenotypic test is an easily applied, rapid, and cost-efficient tool for identification of *Actinomyces* species. The classical method is based on phenotypic tests such as gas production from glucose, urease, catalase, and acid production [8]. However, it is difficult to determine the taxon when species have the same characteristics [4]. With the low cost of 16S rRNA sequencing in recent years, it is possible to use this reliable and accurate approach currently available in clinical practice. In this study, we used 16S sequencing to confirm the pathogen, which was identified as *Actinomyces viscosus*.

In general, *Actinomyces* species are susceptible to penicillin and beta-lactam antibiotics [54]. Some studies reported that actinomycosis can be treated successfully with ceftriaxone [23,27,54], but several case reports noted that some isolates were resistant to ceftriaxone, such as *A. europaeus* and *A. graevenitzii* [12,54]. A retrospective assessment of antimicrobial susceptibility testing of isolates showed that the *Actinomyces* spp. were also susceptible to carbapenems, tetracyclines, clindamycin, and vancomycin as alternative treatments [54]. However, another interesting observation was that *Actinomyces* isolates were resistant to doxycycline [12] and clindamycin, showing a high range from 30% to 80% with poor susceptibility rates [55]. Additionally, in the present study, it was found that *A. meyeri* and *A. odontolyticus* were resistant to tetracycline and vancomycin; in most *Actinomyces* spp., isolates showed high resistance to metronidazole and quinolones as “intrinsic resistance” [55]. Therefore, the susceptibility profiles of *Actinomyces* are important as they help to inform the selection of an appropriate treatment [54]. In our study, the patient recovered well after intravenous ampicillin treatment.

A recent study on the healthy human oral microbiome showed that *Actinomyces* species are part of the oral flora in Taiwanese populations [56]. Human isolates of *Actinomyces viscosus* showed high phenotypic and serological relatedness to *A. naeslundii* [57]. Therefore, we used 16S rRNA sequencing analysis for identification of *A. viscosus*; it was also possible to differentiate it from other closely related species in the genera *Arcanobacterium* and *Actinobaculum* [58,59]. The oral cavity could be considered a causative factor in human actinomycosis [60]. The isolation of *Actinomyces* spp. from blood culture is traditionally regarded as clinically significant [61]. However, the possibility of contaminants does exist [62]. Sound clinical judgment based on the presence of risk factors is essential to interpretate the results. According to the literature review of original clinical studies on *Actinomyces*, this species can become pathological when superadded by periodontal disease and poor oral hygiene, leading to the development of infections. The mucosal barrier is disrupted by triggering factors such as plaque, tooth cavities, and periodontitis in the case of oral infections [19,63,64].

In this study, our patient had poor oral hygiene and chronic periodontitis, which were considered risk factors. *A. viscosus* is an opportunistic pathogen in the oral cavity. This may have been the source of bacteremia in our patient, who had hematogenous spreading, which led to cellulitis of the right lower leg. Furthermore, immunocompromised patients, such as those with diabetes, human immunodeficiency virus (HIV), etc., patients who have undergone a surgical or invasive procedure, and patients with local tissue damage, are also traditionally considered to be at greater risk for actinomycosis.

## 5. Conclusions

Actinomycosis is considered a curable disease that can be easily treated with penicillin and amoxicillin as the first-line treatment. However, chronic granulomatous clinical presentation, selective cultivation of an environment, and species differences in susceptibility profiles may result in clinical confusion, which may lead to treatment failure. In summary, we describe a case with fever and progressive cellulitis. The *Actinomyces* species was isolated from blood culture, which was further identified as *A. viscosus* by 16S rRNA sequencing. The cellulitis improved after pathogen-directed antibiotics. Evaluation of the risk factors of a patient with *Actinomyces* bacteremia and further identification of *Actinomyces* species may increase the likelihood of successful treatment.

## Figures and Tables

**Table 1 medicina-57-01064-t001:** Characteristics and identification profile of *Actinomyces* bacteremia cases reported from 2010 to 2020.

(**A**)								
**Year**	**Age/Sex**	**Nation**	**Underlying Disease**	**Clinical** **Presentation**	**Diagnosis**	**Treatment**	**Outcome**	**Ref.**
2011	51/M	South Korea	Alcoholic liver cirrhosis	Fever (37.9 °C) with hematemesis	Procedure-related bacteremia	Variceal ligation Cefpiramide	Death	[22]
2012	67/M	South Korea	Hepatitis B virus infection	Fever Cough Purulent sputum Headache	Lung abscesses	Ceftriaxone	Recovered	[23]
2013	40/F	Belgium	Crohn’s disease	Fever (40.7 °C) Shivering Vomiting after IVF	Pelvic inflammatory disease with abscess	Amoxicillin–clavulanic acid Exploratory laparotomy	Recovered	[24]
2014	31/F	USA	Multiparous IUD placement	Fever (38.9 °C) Pelvic pain for weeks	Tubo-ovarian abscesses	Penicillin G	Recovered	[23]
2014	26/M	USA	Recent right partial orchiectomy for epidermoid cyst	Fever (39.5 °C) Right testis swelling tenderness	Testicular abscess	Piperacillin/tazobactam and vancomycin IV	Recovered	[25]
2014	90/F	Japan	Diabetes mellitus Hypertension	Deteriorated mental status	*Actinomyces meyeri* meningitis	Ampicillin, ceftriaxone, ceftazidime, vancomycin Acyclovir	Did not regain consciousness	[26]
2014	59/M	Croatia	Ulcerative colitis Colonoscopy (2 months ago)	Fever (39.9 °C) Vomiting Watery stools Abdominal pain	Abdominal actinomycosis	Ciprofloxacin with metronidazole Ceftriaxone	Recovered	[27]
2015	80/F	Japan	Bedridden	Fever Impaired consciousness	Pyometra	Ampicillin–sulbactam	Recovered	[28]
2015	53/M	Denmark	Diabetes mellitus Recurrent skin abscesses COPD Obesity	Fever Painful swelling of right breast	Breast abscess	Dicloxacillin	Recovered	[29]
2016	47/M	Thailand	Not mentioned	Cough and chest congestion	Aspiration pneumonia	Vancomycin and piperacillin–tazobactam	Recovered	[30]
2016	50/F	USA	Hypertension	Weakness Nausea Vomiting Diarrhea	Streptococcal toxic shock syndrome Pelvic actinomycosis	Clindamycin, ampicillin–sulbactam	Recovered	[31]
2018	23 weeks gestational age/F	USA	(Mother) severe HELLP syndrome Cesarean section	Prematurity	Neonatal sepsis	Ampicillin Penicillin	Discharge to another healthcare facility	[32]
2018	56/F	USA	Repaired TOF	Fever Epigastric pain and melena	Septic pylephlebitis	Penicillin G Ertapenem	Recovered	[33]
2019	61/M	USA	Endocarditis, ESRD under H/D Atrial fibrillation Monoclonal gammopathy	Fever (39.4 °C) Confusion Weakness Slurred speech after H/D	Infective endocarditis by *A. neuii* Aortic root abscess and presumed cerebral septic emboli	Surgery Ampicillin	Recovered	[34]
2019	84/F	USA	DM HTN Anemia CAD Frequent UTI due to incontinence	Severe right thigh pain	Necrotizing soft-tissue infection	Debridement Clindamycin Vancomycin Piperacillin–tazobactam	Recovered	[35]
2020	60/M	Denmark	DM	Urinary retention Macroscopic hematuria	UTI	Mecillinam	Recovered	[36]
2020	52/F	Philippines	UTI Vaginitis Allergic to Penicillin	Lower abdominal and flank pain	Urosepsis with shock *A. turicensis* bacteremia	Ceftriaxone	Recovered	[37]
2020	8 months old/F	France	Metastatic neuroblastoma under chemotherapy	Fever and neutropenia	Neutropenic fever	Imipenem	Recovered	[38]
Abbreviation: IVF = in vitro fertilization; IUD = intrauterine device; COPD = chronic obstructive pulmonary disease; HELLP = hemolysis, elevated liver enzymes, and low platelets; TOF = tetralogy of Fallot; ESRD = end-stage renal disease; H/D = hemodialysis; DM = diabetes mellitus; HTN = hypertension; CAD = coronary artery disease; UTI = urinary tract infection; TVOR=Trans-vaginal oocyte retrieval.								
(**B**)								
**Year**	**Age/Sex**	**Nation**	**Diagnosis**	**Isolated Species**	**Identified Method**	**Treatment**	**Previous** **Invasive** **Procedure**	
2011	51/M	South Korea	Procedure-related bacteremia	*A. graevenitzii*	16S rRNA	Varice ligation Cefpiramide	Yes	
2012	67/M	South Korea	Lung abscesses	*A. cardiffensis*	16S rRNA	Ceftriaxone	No	
2013	40/F	Belgium	Pelvic inflammatory disease with abscess	*A. urogenitalis*	MALDI-TOF MS + 16S rRNA	Amoxicillin–clavulanic acid Exploratory laparotomy	Yes	
2014	31/F	USA	Tubo-ovarian abscesses	*A. naeslundii.*	16S rRNA	Penicillin G	Yes (IUD)	
2014	26/M	USA	Testicular abscess	*A. neuii. (blood and abscess)*	Not mentioned	Piperacillin/tazobactam Vancomycin IV	Yes	
2014	90/F	Japan	*Actinomyces meyeri* meningitis	*A. meyeri.*	RapID ANA II	Ampicillin Ceftriaxone Ceftazidime Vancomycin Acyclovir	No	
2014	59/M	Croatia	Abdominal actinomycosis	*P. aeruginosa* *A. naeslundii*	Not mentioned	Ciprofloxacin with metronidazole Ceftriaxone	Yes	
2015	80/F	Japan	Pyometra	*A. turicensis* *Clostridium clostridioforme*	MALDI-TOF MS + 16S rRNA	Ampicillin–sulbactam	No	
2015	53/M	Denmark	Breast abscess	*A. europaeus* (Blood and abscess cavity)	MALDI-TOF MS + 16S rRNA	Dicloxacillin	No	
2016	47/M	Thailand	Aspiration pneumonia	*A. radicidentis*	16S rRNA + Retrospective MALDI-TOF MS	Vancomycin Piperacillin–tazobactam	Yes (IUD)	
2016	50/F	USA	Streptococcal toxic shock syndrome Pelvic actinomycosis	*A. odontolyticus*	Not mentioned	Clindamycin Ampicillin–sulbactam	No	
2018	23 weeks gestational age/F	USA	Neonatal sepsis with bacteremia	*A. viscosus*	16S rRNA.	Ampicillin Penicillin	No	
2018	56/F	USA	Septic pylephlebitis secondary to *Actinomyces* bacteremia	*A. meyeri*	Biochemical analysis	Penicillin G Ertapenem	No	
2019	61/M	USA	Infective endocarditis by *A. neuii.* Aortic root abscess and presumed cerebral septic emboli	*A. neuii*	MALDI-TOF MS	Surgery Ampicillin	No	
2019	84/F	USA	Necrotizing soft-tissue infection	*A. europaeus* *A. schaalii*	Biochemical analysis	Debridement Clindamycin Vancomycin Piperacillin–tazobactam	No	
2020	60/M	Denmark	UTI	*A. urogenitalis*	MALDI-TOF MS	Mecillinam	No	
2020	52/F	Philippines	Urosepsis with shock	*A. turicensis*	Biochemical analysis	Ceftriaxone	No	
2020	8 months old/F	France	Neutropenic fever	*Tsukamurella pulmonis* *(Actinomycetales)*	MALDI-TOF MS 16s+ secA1 Gene Sequencing	Imipenem	Yes	
Abbreviation: IUD = intrauterine device; MALDI-TOF MS = matrix-assisted laser desorption ionization time-of-flight mass spectrometry; UTI = urinary tract infection.								

**Table 2 medicina-57-01064-t002:** Characteristics of *Actinomyces viscosus* infection published from 1977 to 2020.

Year	Age/Sex	Nation	Underlying Disease	Clinical Presentation	Specimen	Diagnosis	Treatment	Outcome	Ref.
1977	62/F	USA	Nil	Submandibular swelling	Ductal discharge	Submandibular abscess	Flucloxacillin	Recovered	[39]
1977	76/F	USA	Multiple myeloma	Fever with crackle	Blood	Pneumonia	Ampicillin Cloxacillin	Recovered	[39,40]
1978	8/M	USA	Nil	Fever and a cervical mass	Tissue	Neck cellulitis	Phenoxymethyl Penicillin	Recovered	
1979	7/F	USA	Fall accident 6 months ago	Enlarging macular lesion on the right lower chest	Chest wall pus	Chest wall abscess with rib involvement	Rib resection Clindamycin	Recovered	[41]
1979	18/M	USA	Nil	Fever (40 °C) Chest pain Cough Hemoptysis	Percutaneous transtracheal aspiration	Pneumonia	Ticarcillin	Recovered	[42]
1979	49/M	USA	Penectomy for carcinoma of the penis	Fever Cough with expectoration Night sweats Chest pain on	Transtracheal aspiration	Pneumonia	Ticarcillin Penicillin	Recovered	
1981	27/M	USA	Alcoholism	Fever (38.2 °C) Cough	Biopsy	Lung abscess	Penicillin G	Recovered	
1981	21/M	USA	Sickle cell disease	Fever (38 °C) with cold sensation Cough	Tissue	Lung abscess	Penicillin	Recovered	[43]
1984	60/M	USA	Nil	Cough with left chest pain	Tissue	Chest wall abscess	Excision Penicillin	Recovered	
1998	55/M	HK	Nil	Epigastric pain and weight loss	Biopsy	Esophageal actinomycosis	Amoxicillin–clavulanate	Recovered	[44]
1998	81/M	USA	Valvular heart disease	Fever (38.5 °C) Depressed mood Suicidal ideation Anorexia Back pain Weight loss	Blood	Endocarditis	Ceftizoxime	Recovered	[45]
1999	78/M	USA	Cataract Hypertension Gout GERD	Severe right eye pain	Vitreous washings	Endophthalmitis	Penicillin	Recovered	[46]
2005	27/F	Italy	Nil	Pain and tenderness in the right breast	Abscess	Breast abscess	Excision Augmentin	Recovered	[47]
2005	43/F	USA	Fever without focus 1 year before	5 days of subjective fever Hemoptysis Productive cough Increased dyspnea on exertion	Aortic valve tissue cytology Previous blood culture	Aortic valve endocarditis	Aortic valve repair Vancomycin, Gentamicin Ceftriaxone	Recovered	[48]
2007	35/F	India	Nil	Fever and throbbing type of pain of back	Pus and biopsy	Mycetoma	Penicillin Cotrimoxazole	Recovered	[49]
2008	17/F	India	ALL under induction chemotherapy	Fever Lung consolidation	Pleural fluid	Empyema	Imipenem–cilastatin	Recovered	[50]
2011	7/F	Tunis	Nil	Fever (39 °C) Vomiting	Purulent liquid from neurosurgical drainage	Subdural empyema	Ampicillin	Death	[51]
2018	74/M	USA	COPD Smoker Psoriatic arthritis on methotrexate	Generalized weakness and difficulty to ambulate	Neck abscess	Neck, lung and brain abscess	Penicillin	Recovered	[10]
2018	23 weeks gestational age/F	USA	(Mother) severe HELLP syndrome Cesarean section	Prematurity	Blood	Neonatal sepsis	Ampicillin Penicillin	Discharge to another healthcare facility	[32]

Abbreviation: GERD = gastroesophageal reflux disease; ALL= acute lymphocytic leukemia; COPD = chronic obstructive pulmonary disease; HELLP = hemolysis, elevated liver enzymes, and low platelets.

## Data Availability

All the data are available from the corresponding author upon reasonable request.

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
