# Peer review of "Clinical Characteristics of Actinomyces viscosus Bacteremia"

_medicina, 2021, doi:10.3390/medicina57101064_

Round 1
Reviewer 1 Report
A nice paper presenting a rare case of bacteremia and celullitis caused with Actinomyces viscosus. Further litetrature overview is useful and informative.
Author Response
Dear reviewer,
Thanks for your encouragement,we will improve the research content by providing more information to the clinic in the future.
Best regards,
Jui-Hsing Wang
Director of Division of Infectious Disease
Division of Infectious Disease, Taichung Tzu Chi Hospital
Reviewer 2 Report
Comments to the Author
Major comments:
- The microbiological part of the introduction is too long.
- The sentences without content in discussion should be excluded.
- Line 93/ incomprehensible statement / …the main source of oral bacteria was actinomycosis.
- Line 194/5/ Actinomyces bacteria are increasingly being isolated from human clinical specimens and are capable of causing disease . Is this true? The selected reference does not confirm this.
- Line 217/8/ It is not true. 16S rRNA sequencing test does not provide useful informations for antimicrobial selection.
- Isolation of Actinomyces spp. from blood culture should raises the question of whether these organisms are blood culture contaminants or represent transitory bacteraemias caused by translocation from commensal sites. Especially with a clinical picture such as cellulitis.This dilemma should be further clarified in the discussion.
Minor comments:
- The data in Table 1 and two should be combined in one table.
- Line 248 / …source of invasive disease.
- In revised manuscript the text should be carefully checked and grammar errors should be corrected.
Author Response
Dear reviewers of Medicina:
First of all, we would like to thanks for editors and reviewers for all the extremely helpful comments provided for our paper. We have taken response to these comments, and hope that a revised version of the manuscript will still be considered by Medicina.
In the response letters below, we’ve addressed all comments from all the suggestions. The original comments are shown in italics, while our responses are shown in bold text. In addition, the changes to the manuscript are shown in red color with highlight. Please see the attachment with the email.
Sincerely,
Jui-Hsing Wang
Director of Division of Infectious Disease
Division of Infectious Disease, Taichung Tzu Chi Hospital